# Changes in Serum Inflammatory Markers and in Clinical Periodontal Condition After Non-Surgical Periodontal Treatment in Hypertensive Patients

**DOI:** 10.3390/biomedicines13020374

**Published:** 2025-02-05

**Authors:** Francina María Escobar Arregoces, Nelly S. Roa, Juliana Velosa-Porras, Lina Velásquez Rodríguez, María José Merchan, Jean Carlos Villamil Poveda, Liliana Otero, Álvaro J. Ruiz, Catalina Latorre Uriza

**Affiliations:** 1Centro de Investigaciones Odontológicas (CIO), Faculty of Dentistry, Pontificia Universidad Javeriana, Bogotá 110311, Colombia; juliana.velosa@javeriana.edu.co (J.V.-P.); jean.villamil@javeriana.edu.co (J.C.V.P.); lotero@javeriana.edu.co (L.O.); clatorre@javeriana.edu.co (C.L.U.); 2Faculty of Dentistry, Pontificia Universidad Javeriana, Bogotá 110311, Colombia; velasquezlina@javeriana.edu.co (L.V.R.); mariamerchan@javeriana.edu.co (M.J.M.); 3Departamento de Medicina Interna, Faculty of Medicine, Pontificia Universidad Javeriana, Bogotá 110311, Colombia; aruiz@javeriana.edu.co; 4Departamento de Epidemiología Clínica y Bioestadística, Faculty of Medicine, Pontificia Universidad Javeriana, Bogotá 110311, Colombia

**Keywords:** hypertension, periodontal disease, periodontitis, periodontal infection, periodontal debridement, inflammation mediators, cytokines

## Abstract

Background: Chronic inflammatory disorders, such as periodontitis, may contribute to pro-hypertensive inflammation. Objectives: This study aimed to analyze changes in parameters for periodontitis, such as periodontal inflamed surface area (PISA) and serum inflammatory markers, following non-surgical periodontal treatment in hypertensive patients. Methods: A quasi-experimental pre-and-post study was conducted, involving 42 controlled hypertensive patients with periodontitis. The patients underwent periodontal assessment and tests, including complete blood count, glucose, triglycerides, HDL-C, LDL-C, and serum levels of inflammatory biomarkers. All patients received scaling and root planning treatment in a single session and were reevaluated one month after the treatment. Results: Post-treatment evaluations showed significant improvements in periodontal inflammation parameters, such as pocket depth, attachment level, bleeding on probing, and biofilm percentage, with statistically significant differences (*p* < 0.001). There were decreases in serum VEGF levels (*p* < 0.001) and reductions in PISA associated with declines in cytokine levels such as IL-10, IL-6, IL-12p70, IL-17A, and VEGF. PISA for IL-6 and IL-10 had a positive correlation before periodontal treatment and with IL-1β and IL-10 after treatment. Conclusions: Hypertensive patients with periodontitis who underwent non-surgical periodontal treatment showed improvements in their periodontal condition, a decrease in cytokine levels such as VEGF, and reductions in PISA associated with declines in cytokines such as IL-10, IL-6, IL-12p70, IL-17A, and VEGF. These findings confirm the role of inflammation in hypertensive patients with periodontitis.

## 1. Introduction

High blood pressure is a chronic non-communicable disease with a significant impact on global mortality. The overall prevalence of hypertension in adults is estimated to be between 30% and 45%. The PURE study found a prevalence of 40.3% in the urban population and 34.9% in the rural population of Colombia. This high prevalence of hypertension is consistent worldwide, regardless of income status, whether in low-, middle-, or high-income countries. When analyzed according to DALYs (disability-adjusted life years) and attributable deaths, high systolic blood pressure (SBP) is the leading cause of disease burden [1,2].

In recent years, there has been an intensified search to identify potential risk factors for hypertension, reduce its prevalence, and mitigate its impact. Clinical evidence suggests a significant role for the systemic inflammatory response in the development of hypertension. Chronic inflammatory disorders could serve as a substrate for pro-hypertensive inflammation [3]. Periodontitis, a chronic inflammatory disease, represents the sixth most prevalent condition globally and has been linked to hypertension [4]. The proposed explanation for this association is based on the idea that periodontitis could influence inflammatory biomarkers, such as TNF-α, Interleukin 1 Beta (IL-1β), and Interleukin 6 (IL-6), which may contribute to increased blood pressure. Treating periodontitis by reducing systemic pro-inflammatory interleukins could potentially help lower blood pressure [5]. However, the evidence remains inconclusive. Some authors support the relationship between periodontal disease and hypertension [6,7], while others have not found a clear link [8].

The effect of closed-field scaling and root planning on systemic inflammation biomarkers in hypertensive patients has been widely studied. Escobar et al. [9] also assessed the effect of non-surgical periodontal treatment on serum inflammation levels in 19 hypertensive patients. All cytokine levels decreased from the initial examination to the reevaluation at one month. The cytokines that showed a statistically significant difference were interleukin-1β and vascular endothelial growth factor (VEGF) (*p* = 0.04 and *p* = 0.004, respectively). Hypertensive patients with periodontitis who underwent non-surgical periodontal treatment showed a decrease in pro-inflammatory cytokine levels, highlighting that non-surgical periodontal treatment reduces systemic pro-inflammatory cytokine levels in controlled hypertensive patients. Based on these findings and to enhance the result, the objective of this study was to analyze the effect of periodontal treatment with scaling and root planning in a closed field on the systemic inflammatory response in a group of patients from the Colombian population, increasing the previous sample size and including periodontal inflamed surface area (PISA). This led to the following research question: What is the effect of closed-field scaling and root planning on serum levels of IL-1β, IL-4, IL-6, IL-8, IL-10, IL-12p70, IL-17A, VEGF, and TNF-α in patients with periodontitis and high blood pressure?

## 2. Materials and Methods

A quasi-experimental before-and-after study was conducted to evaluate changes in systemic inflammatory biomarkers before and after closed-field scaling and root planning. This study included patients over 30 years of age with periodontitis stages II, III, and IV, Grades A and B, controlled hypertension, and at least six teeth present in the mouth. Patients with diabetes, smokers, those on antibiotic treatment in the last three months, those who had received periodontal treatment in the last six months, and patients with immunological diseases that could alter blood pressure, such as rheumatoid arthritis and systemic lupus erythematosus, were excluded.

This research was conducted with the approval of the Research and Ethics Committee of the Faculty of Dentistry at Pontificia Universidad Javeriana (CIEFOPUJ) under the number 007/2015. All included patients signed an informed consent form to participate in the study, in accordance with the 1964 Helsinki Declaration [10] and its subsequent amendments. We confirm that all subjects received clear information and gave their consent prior to participating in the study.

Collection of Clinical Information: An initial periodontal examination was conducted, considering the presence of gingival bleeding, attachment level, and probing depth, as well as the periodontal inflamed surface area (PISA). For probing depth, six measurements were taken from all teeth, on the buccal side (mesial, middle, and distal) and the lingual side (mesial, middle, and distal). The measurement from the gingival margin to the bottom of the groove or pocket was recorded as the sulcus or periodontal pocket. Measurements up to 3 mm were considered normal, while those from 4 mm onwards were classified as periodontal pockets.

The measurement of the gingival margin was taken from the cementoenamel junction to the gingival margin. With these sulcus depth and margin data, attachment levels were calculated, and the periodontal diagnosis was established according to Caton in 2018. The periodontal assessment was conducted by two calibrated periodontists using a North Carolina-type graduated manual probe; the periodontists had more than 20 years of clinical experience. According to the study protocol, biofilm control was measured using O’Leary’s index, and oral hygiene instruction was provided.

Laboratory exams were performed at the Hospital Universitario San Ignacio (HUSI), with patients fasting. Tests included a complete blood count, glucose, triglycerides, high-density lipoprotein cholesterol (HDL-C), low-density lipoprotein cholesterol (LDL-C), and serum levels of inflammatory biomarkers, including IL-1β, IL-4, IL-6, IL-8, IL-10, IL-12p70, IL-17A, VEGF, and TNF-α. After the patients had breakfast and rested for 1 h, a professional from the Cardiology department at HUSI measured their blood pressure.

All patients were scheduled for periodontal treatment. The treatment was carried out according to the European Federation of Periodontology (EFP) clinical practice guideline (CPG) for the treatment of periodontitis. The first step of the therapy aimed to guide behavior change by motivating the patient to successfully remove the supragingival dental biofilm and control risk factors. The second step of the therapy (cause-related therapy) aimed to control (reduce/remove) the subgingival biofilm and calculus through subgingival instrumentation [11]. A periodontal reevaluation was conducted four to five weeks after each patient’s treatment, and the same baseline values from the initial examination were recorded. On the same date, prior to the periodontal reevaluation, a second blood sample was taken to measure glucose, cholesterol, triglycerides, HDL-C, LDL-C, systemic inflammatory biomarkers, and complete blood count. A second blood pressure measurement was also taken under the same conditions as the first one.

Two blood samples were taken: one in a dry tube and another with an anticoagulant. The sample obtained in the anticoagulant tube was processed in the clinical laboratory at HUSI to analyze complete blood count, glucose, total cholesterol, triglycerides, HDL-C, and LDL-C. From the dry tube sample, blood serum was obtained after centrifuging at 10,000 rpm for 10 min; the supernatant was collected and frozen in 1.5 mL vials at −20 °C until used to measure pro-inflammatory cytokines at the CIO.

Measurement of Blood Pressure: To establish blood pressure, measurements were taken from both arms with an interval of at least two minutes. Patients had to meet the following requirements for an accurate blood pressure reading: the measurement was performed after 10 min of rest, and patients should not have consumed caffeine within 30 min before the blood pressure reading. The patient should be seated with their back supported by the chair, legs touching the floor, arm raised, relaxed, and not clenched. The cuff was placed in contact with the skin, with the arm supported and at heart level. Patients were instructed not to talk during the measurement. The measurement was performed on both arms, and the same calibrated electronic blood pressure monitor was used for all patients.

Measurement of Serum Levels of Inflammatory Biomarkers: Serum cytokines were quantified using the Luminex system with the Milliplex Inflammation Human Cytokine kit. The test was performed on a plate where reagents and working standards were prepared according to the manufacturer’s instructions as described above [9]. All samples were prepared in duplicate.

Statistical Analysis: The demographic characteristics of the included patients, as well as the results of the periodontal evaluation, blood pressure, and systemic inflammatory biomarkers, are reported as means, medians, ranges, standard deviations, and 95% confidence intervals. Comparisons between groups were made using Student’s *t*-test or chi-square test, as appropriate. Additionally, the Kruskal–Wallis test and the Wilcoxon test were used to evaluate changes in cytokine levels. For this study, a *p*-value < 0.05 (two-tailed) was considered statistically significant.

## 3. Results

The study included a total sample of 44 patients, of whom 3 were excluded due to incomplete follow-up. From the final sample, 60.9% were women (25 patients) and 39.1% were men (16 patients), with an average age of 57.8 years 95% CI (54.8; 60.8) (Table 1).

Of the 41 patients included in the study, the periodontal condition was assessed according to Caton 2018 [12] at the beginning of the study. Eleven patients were diagnosed with periodontitis stage III, Grade A. Of these, only one was diagnosed with a reduced but healthy periodontium at the time of reevaluation, while three were diagnosed with biofilm-induced gingivitis on a reduced periodontium in periodontally treated patients. Fourteen patients were initially diagnosed with periodontitis stage III, Grade B, and of these, only one was diagnosed with periodontitis stage III, Grade A at reevaluation, and one with a reduced but healthy periodontium. Five patients were diagnosed with periodontitis stage IV, Grade A at the start, and only one was later diagnosed with biofilm-induced gingivitis on a reduced periodontium after periodontal treatment. Eleven patients were diagnosed with periodontitis stage IV, Grade B at both the beginning of the study and the time of reevaluation (Table 2).

On average, 22.7 teeth per patient were evaluated. At the beginning of the study, an average of 14.5 teeth per patient had periodontitis. The average pocket depth was 3.9 mm, with a range of 3.5 mm to 4.2 mm pre-treatment. At the one-month reevaluation, the average number of teeth was 22.6, of which 8.8 had periodontitis. The average pocket depth was 3.19 mm, with a range of 2.9 mm to 3.4 mm. The clinical attachment level was 3.42 mm, bleeding on probing was 33.1%, and the plaque index was 34.9%. There was also an improvement in PISA and clinical parameters one month after reevaluation (Table 3).

The average blood pressure at the beginning of the study was 131.1/81.8 mm Hg, which changed to an average of 126.5/79.1 mm Hg at the end of the treatment. Although this change was statistically significant, it does not reach clinical relevance. (Table 4).

Although the clinical laboratory results show a decrease in all evaluated parameters at the time of reevaluation, only the triglycerides and HDL showed statistically significant differences (Table 5).

Changes in the levels of cytokines evaluated in the patient sample were observed at the time of reevaluation through Student’s *t*-test and the Wilcoxon test. Only VEGF showed decreased levels with statistically significant differences in the Wilcoxon test and a high tendency to present high levels of IL-17. Additionally, it is evident that TNF-α remains the same before and after treatment, and for most subjects, its expression is above the cut-off point. Pro-inflammatory cytokines such as IL-1β, IL-6, and IL-8 are below the cut-off point before and after treatment (Table 6 and Figure 1).

When categorizing the periodontal diagnosis into periodontally healthy and periodontitis groups, no statistically significant difference was observed in these pro-inflammatory markers at the one-month reevaluation (Table 7 and Figure 2), despite observing changes in periodontal diagnosis after treatment (Table 2)

According to the behavior of cytokines after treatment, although there were no significant differences, it was observed that for most of them, especially IL-10 and IL-1β, the mean was below the cut-off point in all periodontal disease diagnoses, except for IL-17, IL-6, and IL-4 in stage III Grade A and stage IV Grade B periodontitis, which remained elevated. Regarding VEGF, its levels remain above the cut-off point even in healthy individuals, with a tendency to be lower in more advanced stages of periodontal disease, such as stage IV Grade A and B, after treatment (Figure 2).

The correlation of cytokines with the inflamed periodontal surface area (PISA) was evaluated before and after periodontal treatment. PISA for IL-6 and IL-10 before treatment (r = 0.5808; r = 0.5740, respectively). PISA after treatment for IL-1β and IL-10 (r = 0.5410 and r = 0.5778, respectively; see Table 8). A reduction in the inflamed periodontal surface area (PISA) was associated with a reduction in the cytokines IL-10, IL-6, IL-12p70, IL-17A, and VEGF after periodontal treatment (Figure 3).

From the articles reviewed, there is no definitive cutoff point for cytokines, as they are detected using different techniques. However, the cutoff point for cytokines in this study was established by taking into account the reported values of IL-1β [14], IL-6, IL-10 [15], IL-8, TNF-α [16], IL-12p70 undetectable in sera from healthy subjects [17], IL-4 [18] VEGF [19], IL-17A [13,20,21]. Statistically significant differences were found between normal and abnormal values before and after treatment in IL-10, IL-6, TNF-α, IL-12p70, IL-17A, and VEGF (Table 8 and Figure 1), indicating that most subjects presented normal serum levels of cytokines IL-10, IL-1β, IL-6, IL-8, IL-12p70, and IL-17A, and elevated levels of TNF-α and VEGF after periodontal treatment (Table 9).

The values in the pre-treatment and post-treatment columns represent the number of subjects with higher or lower serum cytokine levels relative to the cutoff point.

Additionally, the levels of pro-inflammatory markers were evaluated according to the cutoff point for each cytokine and the inflamed periodontal surface area (PISA). It was found that before periodontal treatment, patients with a larger inflamed periodontal surface area had elevated levels of IL-6, unlike what was observed with VEGF, where patients with a smaller inflamed periodontal surface area showed increased levels. This difference was statistically significant for both cytokines. One month after treatment (reevaluation), significant differences were observed only in the increased inflamed periodontal surface area and elevated levels of IL-8. When evaluating the effect of periodontal treatment on periodontal inflammation and cytokine levels according to the cutoff point, a statistically significant reduction in the inflamed surface area and in the levels of the cytokines was found in IL-10, IL-6, IL-12p70, IL-17A, and VEGF (Table 10).

## 4. Discussion

The effect of periodontal treatment on the systemic inflammatory response in hypertensive patients is a topic of great public health interest. This research aimed to analyze whether non-surgical periodontal treatment improved systemic inflammation biomarkers in hypertensive patients medicated with antihypertensives, increasing the sample size and including parameters for periodontitis such as PISA as a marker of local inflammation progression, based on previous studies [9].

Closed-field scaling and root planning treatment in patients with periodontitis and controlled hypertension resulted in improvement in all clinical parameters, including periodontal pocket depth (PPD), clinical attachment level (CAL), bleeding on probing (BOP), and biofilm index (PI), one month after the treatment, with statistically significant differences (*p* < 0.001). These results are consistent with those previously presented by Albush M and colleagues in 2013. In their study of 40 patients, they compared the effects of surgical and non-surgical periodontal treatment and reported significant decreases in PI, GI, BOP, and PPD after 6 weeks of periodontal scaling and root planning in both groups. Similarly to the present study, the subjects were medicated with various antihypertensive agents, which were thought to potentially influence the periodontal disease process. In previous studies by Albush and colleagues in 2011, hypertensive patients on antihypertensive agents were compared with normotensive individuals, showing higher values in PPD and clinical attachment loss [22].

The average blood pressure at the beginning of the study was 131.1/81.8 mm Hg, which decreased to an average of 126.5/79.1 mm Hg at the end of the treatment. Although this change was statistically significant, it did not reach clinical relevance. Contrary to what was reported in this study, Soares Rodríguez et al., in a study conducted on patients with refractory hypertension, observed a significant improvement in the percentages of PI, BOP, PD, and CAL, which were statistically significant at 90 and 180 days, as well as a significant reduction in CRP blood levels. However, they did not observe a significant reduction in blood pressure parameters during the evaluated follow-up periods [23]. Muñoz et al., in a meta-analysis, showed that only 5 out of 12 intervention studies confirmed a reduction in blood pressure (BP) after periodontal treatment, with reductions ranging between 3 and 12.5 mm Hg in systolic blood pressure (SBP) and between 0 and 10 mm Hg in diastolic blood pressure (DBP) [24]. It could be considered that the treatment of periodontitis may represent a new non-pharmacological therapy to prevent or help control hypertension [24].

Looking more closely at the potential long-term cardiovascular benefits of periodontal therapy in hypertensive patients, it is important to mention Luo et al., who in a systematic review on the effect of periodontal treatments on blood pressure, report that randomized controlled trials (RCTs) have shown that periodontal treatment can significantly reduce plasma levels of systemic inflammatory markers (hs-CRP, IL-6, and fibrinogen) in people with hypertension and periodontitis. Additionally, periodontal therapy significantly improves flow-mediated dilation of the brachial artery, indicating a decrease in endothelial dysfunction. Both hypertension and periodontal infections are common medical conditions. Long-term control of blood pressure in people with hypertension is critically important to prevent associated cardiovascular complications [25].

Regarding the blood levels evaluated in terms of glucose, cholesterol, triglycerides, HDL-C, and LDL-C, a decrease in all evaluated parameters was observed one month after periodontal treatment. The most notable changes were in triglycerides, which decreased by an average of 12 mg/dL, showing a statistically significant decrease after treatment. However, HDL cholesterol (atherosclerosis-protective) also showed a statistically significant decrease. The clinical implications of these results should be evaluated in future studies.

The effect of non-surgical periodontal treatment on triglycerides and glucose is noteworthy, as abnormal glucose and lipid metabolism is very common in people with hypertension. It is well established that elevated triglycerides and glucose increase the risk of cerebrovascular events in the general population. Zegui Huang and colleagues conducted a prospective study on 19,924 hypertensive patients from the Kailuan Study, indicating that a long-term elevated triglyceride/glucose index in hypertensive patients is associated with an increased risk of cerebrovascular events, especially ischemic cerebrovascular attacks. This finding suggests that regular monitoring of the triglyceride/glucose index may help identify individuals at higher risk of suffering cerebrovascular events among patients with hypertension [26].

Reducing inflammation caused by periodontal disease appears to be a key factor in managing patients with hypertension. In this regard, Piotr Szczepaniak and colleagues conducted a reflective analysis on the mechanisms of this link, noting that the critical components of the immune and inflammatory pathogenesis of periodontitis overlap considerably with the immune mechanisms of hypertension. Systemic inflammatory cytokines and CRP serve as biomarkers linking periodontitis and hypertension. They suggest that CRP and white blood cell counts are partially mediated by the association between periodontitis and hypertension. Clinical studies support that both C-reactive protein (CRP) levels and white blood cell (WBC) counts mediate the relationship between periodontal disease and high blood pressure. In particular, the activation of Th1, Th17, regulatory T cells, and pro-inflammatory monocytes is essential in both conditions [27].

It is evident that the immune system plays a critical role in the development of hypertension and in causing damage to target organs. Periodontal disease increases the immune response due to the rise in bacterial load, primarily anaerobic bacteria. Both adaptive and innate immune responses have been implicated in the pathogenesis of primary and secondary forms of hypertension.

Epidemiologically, periodontal disease is one of the most common examples of prolonged inflammation. In periodontitis, gum bacteria activate both innate and adaptive immunity, increasing the levels of pro-inflammatory cytokines circulating systemically. Non-surgical periodontal treatment reduces the bacterial burden and, in turn, the local and systemic inflammatory response. In this context, Czesnikiewicz-Guzik and colleagues emphasized, based on their animal study results, that the systemic activation of T cells, a characteristic of hypertension, was exacerbated by antigenic stimulus from *P. gingivalis*, a periodontopathogenic bacterium. This resulted in an increase in aortic vascular inflammation with an increased infiltration of leukocytes, particularly T cells and macrophages. The expression of Th1 cytokines, IFN-γ and TNF-α, as well as the transcription factor TBX21, increased in the aortas of mice immunized with *P. gingivalis*/IL-12/aluminum oxide, while the IL-4 and TGF-β levels remained unchanged [28]. These findings support the observation of increased levels of TNF-α, which do not change before or after treatment in controlled hypertensive patients (Figure 1). This expression can be explained as a reflection of the hypertension itself and may be independent of periodontal disease, as it is also increased in H/G subjects (Figure 2). Agita et al. reported the role of pro-inflammatory cytokines as TNF-α in the pathogenesis of hypertension, supported by studies designed to test the inhibitory effect of cytokines on blood pressure, by using Etanercept, which, when injected into body to reduce biological activity of TNF-α in mice, reduces the development of hypertension [29].

When analyzing the periodontal condition and cytokines after treatment in periodontally healthy patients and those with periodontitis, although no statistically significant differences were observed in these biomarkers, lower levels of inflammatory biomarkers such as IL-1β, IL-6, IL-8, IL-4, IL-12p70, and IL-17 were evident in patients who were periodontally healthy. For IL-10 and IL-1β, the mean was below the cut-off point in all periodontal disease diagnoses. However, IL-17, IL-6, and IL-4 in stage III Grade A and stage IV Grade B periodontitis remained elevated. This suggests that evaluating systemic cytokines one month after treatment may be too short a period to observe reduced levels of these cytokines.

D’Isidoro et al., in a systematic review, analyzed the effects of non-surgical periodontal treatment on biomarkers of cardiovascular diseases (CVD) to clarify the impact of periodontal disease on systemic inflammation. They reported that non-surgical periodontal therapy significantly reduced systemic inflammation markers, decreased vascular risk, and lowered the possibility of developing cardiovascular disease [30].

Regarding vascular endothelial growth factor (VEGF), which was reduced after periodontal treatment, Touyz et al. analyzed recent advances in hypertension and cardiovascular toxicity following VEGF inhibition, highlighting that inhibiting this biomarker is associated with a higher incidence of cardiovascular diseases, including hypertension, ischemic heart disease, heart failure, QT interval prolongation, and thromboembolism. The magnitude of hypertension induced by this biomarker is significant, and almost all studies report blood pressure (BP) elevation above 150/100 mm Hg. The development of hypertension depends on the VEGF dose [31]. These findings are consistent with our observations, since VEGF levels remained above the cut-off point even in periodontally healthy subjects (Figure 2), with a tendency to be lower in more advanced stages of periodontal disease such as stage IV Grade A and B after treatment. This supports the conclusion that the high presence of systemic VEGF in the patients studied occurs as a consequence of hypertension and is independent of periodontal disease; however, its behavior can be altered by periodontal treatment.

A reduction in VEGF could have a positive impact on systemic health and hypertension management. To emphasize this point, we can mention that VEGF has also been linked to pulmonary arterial hypertension (PAH), which is a relatively common complication in patients with congenital heart disease (CHD). Increased proliferation and migration of pulmonary vascular smooth muscle cells are considered pathophysiological cornerstones in all forms of PAH. Furthermore, neurohormonal activation and endothelial dysfunction are also important pathogenetic features in CHD-associated PAH (CHD-PAH). Giannakoulas et al. performed a systematic review on blood biomarkers and their possible role in CHD-associated pulmonary arterial hypertension. They reported that VEGF was a potent cell mitogen serving as an index of endothelial damage and dysfunction. VEGF histological expression in small pulmonary vessels was increased in CHD–PAH, indicating the underlying intimal proliferation and neoangiogenesis. In addition, vascular expression of VEGF, as assessed immunohistochemically, was higher in patients with severe pulmonary vascular disease persisting after corrective surgery. VEGF expression was elevated in arterial cells of the characteristic lesions of advanced pulmonary plexogenic arteriopathy, suggesting a possible role of proapoptotic factors in the development of irreversible pulmonary vascular changes [32].

When evaluating inflammation biomarkers according to cytokine cutoff points, statistically significant differences were found between normal and abnormal values before and after treatment in IL-10, IL-6, TNF-α, IL-12p70, IL-17A, and VEGF. These results align with those of Montenegro et al., who assessed the effect of periodontal therapy on cardiovascular blood biomarkers. They conducted a randomized clinical trial with single blinding, where the test group (TG) received, as in this research, non-surgical periodontal treatment, while the control group (CG) received a plaque removal session. Plasma levels of C-reactive protein (CRP), glycated hemoglobin, lipids, and cytokines (IL-1β, IL-6, IL-8, IL-10, IFN-γ, and TNF-α) were measured at the start and after three months. In patients with CRP ≥ 3 mg/L, they reported a significant reduction in CRP only in the test group. IL-6 and IL-8 levels were significantly lower at 3 months in patients who received non-surgical periodontal treatment [33].

Vicharenko and Rozhko also analyzed levels of interleukin-6 and tumor necrosis factor alpha (TNF-α) in patients with hypertension and periodontitis, considering standard treatment plus chlorhexidine or the same treatment plus medications (Ca-D3 NICOMED, 2 tablets per day; electrophoresis with “Calcium Gluconate” for 10 sessions; Pentoxifylline, 1 tablet three times a day). Their results showed that after periodontitis treatment in hypertensive patients, the indices of pro-inflammatory cytokines TNF-α and IL-6 decreased and were within normal limits. In patients of the second group (who received the offered treatment scheme including medications), the indices of pro-inflammatory cytokines were significantly lower than in patients receiving the standard treatment scheme [34]. As explained in this background, our study found that the pro-inflammatory cytokines expected to be elevated, such as IL-1B, IL-17A, and IL-8, were often below the cut-off point. This effect can be attributed to the treatment received by the controlled hypertensive patients included. For this reason, the changes in cytokine expression after periodontal treatment were not very noticeable.

The clinical implications of these biochemical changes in interleukins and their relationship with hypertension remain unclear, but it is important to note that recent scientific evidence has consistently highlighted the impact of IL-6 on arterial hypertension. Caiazzo et al. performed a systematic review and meta-analysis, reporting that higher levels of IL-6 might be associated with a greater, but not robust (lower 95% CI = 1.0), risk of developing hypertension. Preclinical evidence suggests that IL-6 might be involved in the initiation, as well as in the progression and maintenance of hypertension through a reduction in nitric oxide bioavailability, an increase in vascular superoxide, regulation of angiotensin II expression, and alterations in vascular function and structure. The research also found that the positive association between IL-6 and the risk of hypertension became insignificant after adjustment for BMI. This finding suggests a link between IL-6 and obesity driving hypertension risk [35].

Regarding the PISA analysis and its relationship with inflammatory cytokines, the results of this study showed a statistically significant reduction in the inflamed periodontal surface area, as well as a reduction in the cytokines IL-10, IL-6, IL-12p70, IL-17A, and VEGF. These results could be related to what was reported by Yvonne Jockel-Schneider et al. [36], who, in a randomized clinical trial with 55 patients, analyzed similar periodontal clinical parameters and the inflamed periodontal surface area before and 12 months after anti-infective periodontal therapy. They indicated that with the reduction in the inflamed periodontal area at twelve months, a significant reduction in pulse-wave velocity (PWV) of −0.6 m/s was observed, along with a better central pulse pressure response (ΔBoP ≥ 88%). They concluded that an effective resolution of the inflamed periodontal surface area could have a beneficial impact on vascular health [36].

It is important to mention that this research focuses on the results obtained at 1 month after periodontal treatment. Recent evidence has reported that in systemically healthy patients, most of the reduction in probing pocket depth (PPD) and gain in clinical attachment level (CAL) occurs within the first 1–2 months after subgingival instrumentation. However, additional benefits in terms of pocket depth reduction occur beyond these early time points. Therefore, further research investigating inflammatory biomarkers at 3 and 6 months after periodontal treatment should be conducted in the future [37].

Based on the results obtained from this work, it is recommended to perform systemic biomarker evaluations over longer periods, for example, at 3, 4, or 6 months after treatment if PISA measures are kept small, to better understand the relationship between periodontal disease and the control of hypertension following non-surgical periodontal treatment.

## 5. Conclusions

Considering the findings of the present study, it can be concluded that:

Non-surgical periodontal treatment in patients with periodontitis and controlled hypertension produced a significant improvement in clinical parameters, including periodontal pocket depth (PPD), clinical attachment level (CAL), bleeding on probing (BOP), and biofilm index (PI), one month after treatment.

The treatment of periodontitis through closed-field scaling and root planning in hypertensive patients reduced both systolic and diastolic blood pressure.

In patients who achieved periodontal health, levels of anti-inflammatory as IL-10 and pro-inflammatory biomarkers IL-6, IL-8, IL-12p70, and IL-17 were reduced, though without statistically significant differences.

Statistically significant differences were found before and after treatment in IL-10, IL-6 at levels that remained normal, and TNF-α that remained with elevated levels without having an effect due to treatment, according to the reference point (normal and abnormal) for these cytokines.

Non-surgical periodontal treatment produced a statistically significant reduction in the inflamed periodontal surface area and in the cytokines IL-10, IL-6, IL-12p70, IL-17A, and VEGF.

## Figures and Tables

**Figure 1 biomedicines-13-00374-f001:**
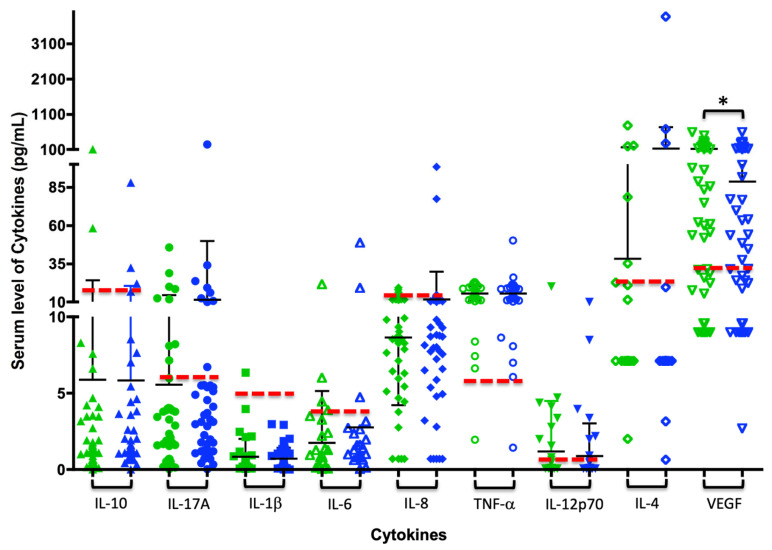
Changes in serum cytokine levels after periodontal treatment: Each figure (triangle, circle, square, or diamond) represents the amount of each cytokine per subject in pg/mL; green indicates the cytokine concentration pre-treatment, and blue indicates post-treatment. The black bars show the average and standard deviation for each analysis group, and the dashed red line illustrates the cutoff point for each cytokine, which determines whether values are above (abnormal) or below (normal). * *p* ≤ 0.05 in the Wilcoxon test.

**Figure 2 biomedicines-13-00374-f002:**
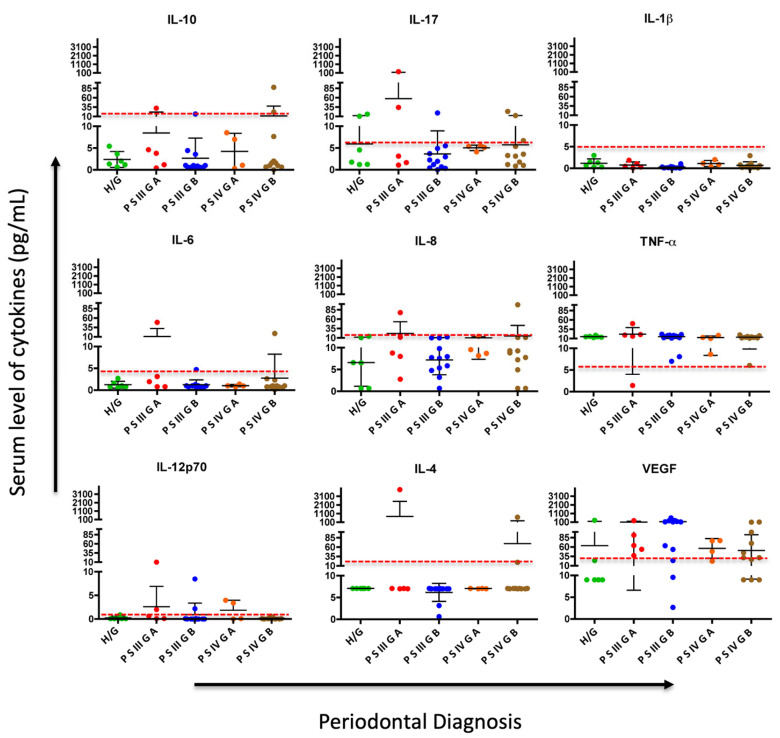
Comparison and dynamics of serum cytokine levels after periodontal treatment and changes in periodontal diagnosis: H/G: Healthy/Gingivitis; PSIIIGA: Periodontitis stage III Grade A, PSIIIGB: Periodontitis stage III Grade B; PSIVGA: Periodontitis stage IV Grade A; PSIVGB: Periodontitis stage IV Grade B. Each point represents the amount of each cytokine per subject in pg/mL, and the dashed red line illustrates the cutoff point for each cytokine, which determines whether values are above (abnormal) or below (normal).

**Figure 3 biomedicines-13-00374-f003:**
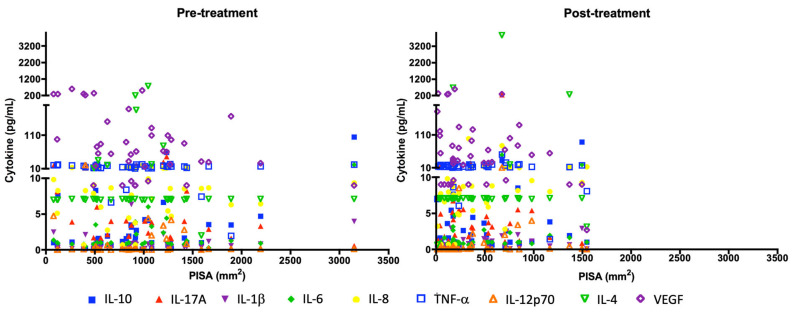
Changes in serum cytokine levels and PISA before and after periodontal treatment. Each symbol represents the amount of each cytokine per subject in pg/mL (y axis) with PISA in mm^2^ (x axis).

**Table 1 biomedicines-13-00374-t001:** Demographic features.

	n	%
Sex		
Female	25	60.9
Male	16	39.1
Total	41	100
	Media	CI 95%
Age	57.8	54.8–60.8

CI 95%: 95% confidence interval.

**Table 2 biomedicines-13-00374-t002:** Periodontal diagnosis (Workshop 2017) [13].

	Pre-Treatment		Post-Treatment	
	n	%	n	%
Reduced healthy periodontium	0	0	2	4.8
Biofilm-induced gingivitis on reduced periodontium in a periodontally treated patient	0	0	4	9.8
Periodontitis stage III Grade A	11	26.8	8	19.5
Periodontitis stage III Grade B	14	34.1	12	29.2
Periodontitis stage IV Grade A	5	12.2	4	9.8
Periodontitis stage IV Grade B	11	26.8	11	26.8
TOTAL	41	100	41	100

**Table 3 biomedicines-13-00374-t003:** Periodontal clinical parameters.

	Pre-Treatment		Post-Treatment		Media Difference	CI 95%	*p* Value
	**Media**	**S.D.**	**Media**	**S.D.**			
Teeth number	22.7	5.65	22.6	5.74	0.02	−0.08–0.13	0.66
Teeth with periodontitis	14.5	6.05	8.80	6.14	5.73	3.99–7.46	0.00
P.D. (mm)	3.90	1.18	3.18	0.88	0.71	0.52–0.91	0.00
CAL (mm)	3.88	1.59	3.42	1.49	0.46	0.30–0.62	0.00
PISA (mm^2^)	956.1	586.5	439.3	412.2	516.7	403.3–630.2	0.00
BOP (%)	58.3	33.4	33.1	23.3	25.1	17.6–32.6	0.00
Biofilm (%)	59.0	25.3	34.9	21.3	24.0	17.5–30.5	0.00

P.D. = probing depth; CAL = clinical attachment level; PISA = periodontal inflamed surface area; BOP = bleeding on probing; S.D. = standard deviation; paired *t*-test. CI 95% = 95% confidence interval.

**Table 4 biomedicines-13-00374-t004:** Blood pressure.

	Pre-Treatment		Post-Treatment				
	**Media**	**S.D.**	**Media**	**S.D.**	**Media** **Difference**	**CI 95%**	***p* Value**
Systolic	131.1	14.7	126.5	14.4	4.62	0.57–8.66	0.02
Dyastolic	81.8	10.1	79.1	9.43	2.74	0.28–5.20	0.02

S.D. = standard deviation; paired *t*-test. CI 95% = 95% confidence interval.

**Table 5 biomedicines-13-00374-t005:** Clinical laboratory results (mg/dL).

	Pre-Treatment		Post-Treatment		Media Difference	CI 95%	*p* Value
	**Media**	**S.D.**	**Media**	**S.D.**			
Glucose	98.0	10.3	96.6	9.92	1.46	−1.20–4.13	0.27
Total cholesterol	193.9	39.1	186.1	34.5	7.75	−1.19–16.7	0.08
Triglycerides	156.3	76.9	144.3	71.1	12.0	−0.17–24.2	0.05
HDL-C	64.1	37.1	45.2	12.1	18.8	7.72–30.0	0.00 *
LDL-C	119.6	33.4	113.2	23.3	6.32	−2.00–14.6	0.13

* S.D. = Standard deviation; paired *t*-test. CI 95% = 95% confidence interval.

**Table 6 biomedicines-13-00374-t006:** Serum cytokines (pg/mL).

	Pre-Treatment		Post-Treatment		Difference		*p* * Value	*p* ** Value
	**Media**	**S.D.**	**Media**	**S.D.**	**Media**	**S.D.**		
IL-1β	0.82	1.16	0.69	0.72	0.13	0.78	0.29	0.79
IL-4	38.2	125.2	124.2	609.6	−85.9	496.8	0.27	0.43
IL-6	1.74	3.40	2.75	7.92	−1.00	6.71	0.34	0.12
IL-8	8.65	4.43	11.6	18.0	−3.02	17.5	0.27	0.87
IL-10	5.88	18.19	5.83	14.57	0.05	9.13	0.97	0.56
IL-12p70	1.16	3.32	0.87	2.13	0.29	2.29	0.42	0.39
IL-17A	5.55	8.78	11.4	38.4	−5.87	38.8	0.33	0.06
TNF-α	15.5	4.71	15.4	7.30	0.14	7.11	0.89	0.12
VEGF	109.9	134.5	88.5	117.2	21.3	141.2	0.33	0.00

* Paired Student test; ** Wilcoxon test.

**Table 7 biomedicines-13-00374-t007:** A comparison of cytokine levels after treatment (pg/mL) according to the periodontal diagnosis. Cytokine levels did not show statistically significant differences with periodontal treatment.

Cytokines	Periodontal Diagnosis		*p* Value *
	Healthy/gingivitis ^Ψ^ Mean (CI 95%)	Periodontitis ^∝^ Mean (CI 95%)	
IL-1β	1.16 (0.32–2.00)	0.61(0.39–0.83)	0.112
IL-4	7.1 (0–¥)	144.2 (−80.8–369.4)	0.718
IL-6	1.25 (0.60–1.90)	3.01 (0.09–5.93)	0.726
IL-8	6.59 (2.11–11.0)	12.5 (5.95–19.1)	0.337
IL-10	2.38 (0.88–3.88)	6.42 (1.05–11.7)	0.605
IL-12p70	0.22 (−0.05–0.50)	0.99 (0.20–1.77)	0.061
IL-17A	5.93 (0.84–11.0)	12.3 (−1.84–26.5)	0.911
TNF-α	15.2 (13.4–16.9)	15.4 (12.7–18.1)	0.631
VEGF	63.7 (−40.8–168.2)	92.8 (52.8–132.8)	0.069

* Kruskal–Wallis test. CI 95% = 95% confidence interval. ^Ψ^ Healthy/Gingivitis includes the diagnoses of: reduced but healthy periodontium and biofilm-induced gingivitis on reduced periodontium. ^∝^ Periodontitis includes the diagnoses of: Periodontitis stage III Grade A and B, and Periodontitis stage IV Grade A and B.

**Table 8 biomedicines-13-00374-t008:** Correlation of cytokines with the inflamed surface area before and after periodontal treatment.

	PISA (A)	VEGF (A)	IL4 (A)	IL12p70 (A)	TNFA (A)	IL8 (A)	IL6 (A)	IL1B (A)	IL17A (A)	IL10 (A)	PISA (B)	VEGF (B)	IL4(B)	IL12p70 (B)	TNFA (B)	IL8 (B)	IL6 (B)	IL1B (B)	IL17A (B)	IL10 (B)
PISA (A)	1.000																			
VEGF (A)	−0.3808	1.000																		
IL4 (A)	0.0178	−0.1620	1.000																	
IL12p70 (A)	−0.1383	0.1231	0.1300	1.000																
TNFA (A)	−0.1044	0.2439	−0.0758	−0.0035	1.000															
IL8 (A)	0.0420	0.2527	0.0079	−0.0010	0.1312	1.000														
IL6 (A)	**0.5808**	−0.1737	0.2225	−0.0243	0.1915	0.0698	1.000													
IL1B (A)	0.2335	−0.1595	−0.0107	0.2207	0.1033	0.0288	0.3753	1.000												
IL17A (A)	−0.0440	−0.0596	0.0123	0.2467	0.1943	0.2706	0.0632	0.1859	1.000											
IL10 (A)	**0.5740**	−0.1538	−0.0324	−0.0475	0.2332	0.1035	**0.8938**	0.4042	0.2949	1.000										
PISA (B)	**0.7955**	−0.3226	0.0382	−0.0401	−0.2603	0.1105	0.3891	0.0529	−0.1686	0.3164	1.000									
VEGF (B)	−0.2308	0.3773	0.3968	0.0854	0.1460	−0.1776	0.0333	−0.2020	−0.1995	−0.1493	0.2487	1.000								
IL4 (B)	0.0122	−0.1455	**0.9198**	0.1465	−0.1324	0.0762	0.2113	−0.0093	−0.0494	−0.0147	0.0974	0.2340	1.000							
IL12p70 (B)	0.0515	−0.0383	0.6349	0.7279	−0.0917	−0.0165	0.0779	0.0913	0.0314	−0.0726	0.1274	0.2032	0.6819	1.000						
TNFA (B)	0.0601	−0.1490	**0.7960**	0.0572	0.3635	−0.0050	0.2338	0.0181	−0.0078	0.0562	0.0316	0.3080	0.7459	0.5056	1.000					
IL8 (B)	−0.0187	−0.1861	0.5616	0.0293	−0.1360	0.2304	0.0756	0.0316	−0.0584	−0.0197	0.1125	−0.0119	0.5749	0.3588	0.3933	1.000				
IL6 (B)	0.2458	−0.1755	**0.8965**	0.1209	−0.0583	0.0518	0.5432	0.1488	−0.0633	0.2991	0.2313	0.2444	**0.9162**	0.6249	0.7420	0.5324	1.000			
IL1B (B)	**0.5410**	−0.3336	0.2579	−0.0144	0.0890	0.0718	0.5101	0.7528	0.1541	0.5086	0.2726	−0.2239	0.2507	0.2216	0.3106	0.2421	0.4246	1.000		
IL17A (B)	0.0190	−0.1473	**0.9502**	0.1661	−0.1078	0.0941	0.1955	0.0435	0.0770	0.0008	0.0647	0.2151	0.9693	0.6859	0.7666	0.5767	**0.9114**	0.3078	1.000	
IL10 (B)	0.5778	−0.1995	0.2635	−0.0241	0.1511	0.0913	**0.9304**	0.4346	0.0087	**0.8674**	0.3855	−0.0783	0.2879	0.1547	0.2684	0.3375	0.5950	0.6227	0.2807	1.000

(A): after; (B): before.

**Table 9 biomedicines-13-00374-t009:** Cytokine analysis according to cutoff points.

	Cutoff Points pg/mL	Pre-Treatment	Post-Treatment	*p* Value
IL-10	≤13.68 >13.68	39 2	37 4	0.000 *
IL-1β	≤5 >5	40 1	41 0	Not calculable
IL-6	≤4.3 >4.3	38 3	38 3	0.000 *
IL-8	≤12.35 >12.35	35 6	35 6	0.161
TNF-α	≤6.11 >6.11	1 40	2 39	0.000 *
IL-4	≤17.76 >17.76	34 7	37 4	0.065
IL-12p70	<1.0 >1.0	33 8	35 6	0.000 *
IL-17A	≤6.13 >6.13	32 9	32 9	0.000 *
VEGF	≤30 >3.0	14 27	16 25	0.002 *

* Fisher’s exact test and Chi-square test. *p* ≤ 0.05.

**Table 10 biomedicines-13-00374-t010:** Comparison of PISA inflamed surface area and cytokine levels according to the cut-off point after treatment (pg/mL).

	Cut-off Points pg/mL	PISA		
		Pre-Treatment	Post-Treatment	*p* Value
IL-10	≤13.68 >13.68	892.7 (CI 95%: 737.4–1048) 2191.5 (CI 95%: −9977.8–14360)	414.4 (CI 95%: 283.8–545) 669.5 (CI 95%: −266.3–1605.5)	0.000 * 0.000 *
*p* value		0.403	0.244	
IL-6	≤4.3 >4.3	888.7 (CI 95%: 729.3–1048) 1809 (CI 95%: −1082.2–4701.2)	411.8 (CI 95%: 284.7–539) 786.8 (CI 95%: −850.6–2424.2)	0.000 * 0.000 *
*p* value		0.007 *	0.131	
IL-8	≤12.35 >12.35	959.9 (CI 95% 747.4–1172.5) 933.6 (CI 95% 528.7–1338.4)	383.6 (CI 95%: 259.1–508) 764.2 (CI 95%: 172.2–1356.1)	0.000 0.1164
*p* value		0.920	0.034 *	
TNF-α	≤6.11 >6.11	No calculable	699 (CI 95%: −5226.4–6624.5) 426 (CI 95%: 294.8–557.1)	
*p* value			0.367	
IL-4	≤17.76 >17.76	966.7 (CI 95%: 744.4–1188.9) 904.5 (CI 95%: 680.7–1128.4)	406.4 (CI 95%: 274.1–538.7) 743.6 (CI 95%: −35.6–1522.9)	0.000 * 0.0630
*p* value		0.667	0.121	
IL-12p70	<1.0 >1.0	953.4 (CI 95%: 734.4–1172) 967 (CI 95%: 573.3–1360.8)	405.8 (CI 95%: 259.9–551.6) 634.8 (CI 95%: 339.4–930.2)	0.000 * 0.000 *
*p* value		0.953	0.212	
IL-17A	≤6.13 >6.13	1013.4 (CI 95%: 795.7–1231) 752.3 (CI 95%: 369.2–1135.5)	450.2 (CI 95%: 299.6–600.9) 400.4 (CI 95%: 82.7–718)	0.000 * 0.000 *
*p* value		0.2430	0.753	
VEGF	≤30 >30	1222.5 (CI 95%: 809.2–1635.7) 817.9 (CI 95%: 634.4–1001.4)	538.5 (CI 95%: 270.7–806.3) 375.8 (CI 95%: 236.1–515.4)	0.000 * 0.000 *
*p* value		0.034 *	0.221	

* *p* ≤ 0.05. Paired *t*-test and *t*-test. CI 95% = 95% confidence interval.

## Data Availability

The data of this research are found in the archives of the Pontificia Universidad Javeriana.

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
