# Peer review of "Changes in Serum Inflammatory Markers and in Clinical Periodontal Condition After Non-Surgical Periodontal Treatment in Hypertensive Patients"

_biomedicines, 2025, doi:10.3390/biomedicines13020374_

Round 1

Reviewer 1 Report

Comments and Suggestions for Authors

Referring to previous research findings, the authors explain the connection between periodontal inflammation and systemic inflammation in hypertension. The authors conducted quasi-experimental, before-and-after design assessed 42 patients, demonstrating significant improvements in clinical periodontal parameters and reductions in pro-inflammatory cytokines, particularly VEGF.  Below are some comments I feel could strengthen the manuscript: 
1.     Multiple cytokines have been analyzed, comprehensive cytokines level determination may provide clear link between hypertension and periodontal (e.g., CRP, IFN-γ, IL-18, etc.).

2.     Line 15, please define the abbreviation “PISA” for first shown.

3.     Line 63, please correct the abbreviation “VEGF”.

4.     Table 1, 3, 4, 5, 7, 10. Please define ‘IC 95%’.

5.     Table 8, please provide high resolution table, line 213.

6. Figure2, if possible, put the cytokines of different conditions in one panel would be helpful for the cytokine levels comparison.

Author Response

Reviewer 1

Comments 1: Multiple cytokines have been analyzed, comprehensive cytokines level determination may provide clear link between hypertension and periodontal (e.g., CRP, IFN-γ, IL-18, etc.).

Response 1: Thank you for pointing this out. We agree with this comment.    

Comments 2:  Line 15, please define the abbreviation “PISA” for first shown

Response 2: Done. The abbreviation “PISA” is defined

Comments 3: Line 63, please correct the abbreviation “VEGF”

Response 3: Done. VEGF abbreviation corrected

Comments 4: Table 1, 3, 4, 5, 7, 10. Please define ‘IC 95%’.

Response 4: Done. IC 95%’ is corrected by CI (Confidence Interval). It is defined that CI 95% = 95% Confidence Interval  in Table 1, 3, 4, 5, 7, 10.

Comments 5 Table 8, please provide high resolution table, line 213.

Response 5:

Comments 6. Figure 2, if possible, put the cytokines of different conditions in one panel would be helpful for the cytokine levels comparison.

Response 6 :

Reviewer 2 Report

Comments and Suggestions for Authors

The manuscript investigates the effects of non-surgical periodontal therapy on inflammatory markers and clinical parameters in hypertensive patients with periodontitis. This topic is clinically significant, given the potential systemic effects of periodontal treatment in individuals with chronic conditions. The study is well-structured, with clear objectives, robust methodology, and a thorough discussion of the findings.

However it might benefit from a few minor changes:

- While the study reports significant changes in VEGF levels and other cytokines, the clinical implications of these biochemical changes remain unclear. Further discussion on the relevance of VEGF reduction in systemic health and hypertension management would add depth to the conclusions.

- The calibration process for periodontists is mentioned but could benefit from additional details regarding inter-examiner reliability to reinforce the robustness of the clinical data.

- The reduction in blood pressure, although statistically significant, was noted as not clinically relevant. A more detailed discussion on the potential long-term cardiovascular benefits of periodontal therapy in hypertensive patients would be valuable.

- Although this study focuses on 1-month outcomes after periodontal treatment, recent evidence highlights that additional clinical improvements may emerge at 3–4 months re-evaluation. Please address to this limitation by citing this article PMID: 38706227

- Please note that according to the latest EFP guidelines, non-surgical periodontal instrumentation is now called step 1 (supragingival and oral hygiene motivation and instructions) and step 2 (subgingival or SRP). Please use this terminology and cite the relevant guideline article PMID: 32383274 

Author Response

Reviewer 2

Comments 1 The manuscript investigates the effects of non-surgical periodontal therapy on inflammatory markers and clinical parameters in hypertensive patients with periodontitis. This topic is clinically significant, given the potential systemic effects of periodontal treatment in individuals with chronic conditions. The study is well-structured, with clear objectives, robust methodology, and a thorough discussion of the findings.

Response 1: Thank you for your positive feedback. We appreciate your recognition of the study's structure and aims.

Comments 2: While the study reports significant changes in VEGF levels and other cytokines, the clinical implications of these biochemical changes remain unclear. Further discussion on the relevance of VEGF reduction in systemic health and hypertension management would add depth to the conclusions.

Response 2: We agree that the clinical implications of these biochemical changes are not fully clear. Limited literature on this topic restricts deeper analysis, but we have attempted to further discuss the relevance of VEGF and IL6 reduction in systemic health and hypertension control.

Comments 3:  The calibration process for periodontists is mentioned but could benefit from additional details regarding inter-examiner reliability to reinforce the robustness of the clinical data.

Response 3: Additional details regarding inter-examiner reliability have been added, clarifying that the periodontists had more than 20 years of clinical experience.

Comments 4: The reduction in blood pressure, although statistically significant, was noted as not clinically relevant. A more detailed discussion on the potential long-term cardiovascular benefits of periodontal therapy in hypertensive patients would be valuable.

Response 4:  We agree with this comment. The discussion has been modified to emphasize the potential long-term cardiovascular benefits of periodontal therapy in hypertensive patients.

Comments 5: Although this study focuses on 1-month outcomes after periodontal treatment, recent evidence highlights that additional clinical improvements may emerge at 3–4 months re-evaluation.

Response 5:  We acknowledge this limitation and have noted that further benefits in terms of pocket depth reduction may be observed beyond the initial time points. This aspect is discussed by citing relevant literature. (PMID: 38706227). It is important to highlight what you mentioned, so recommendations were made for future studies to evaluate, over a period of 3 to 4 months, the re-evaluation, clinical changes, and expression of systemic cytokines while maintaining oral health and expecting decreased PISA values.

Comments 6:  Please note that according to the latest EFP guidelines, non-surgical periodontal instrumentation is now called step 1 (supragingival and oral hygiene motivation and instructions) and step 2 (subgingival or SRP). Please use this terminology and cite the relevant guideline article PMID: 32383274 

Response 6: We have aligned the terms with the EFP guidelines, referring to the treatment steps as step 1 and step 2, and cited the relevant guideline article as suggested. (PMID: 32383274)
